# Effects of Sludge Concentration and Disintegration/Solubilization Pretreatment Methods on Increasing Anaerobic Biodegradation Efficiency and Biogas Production

Jeong-Yoon Ahn [ID] and Soon-Woong Chang *

Department of Environmental Energy Engineering, College of Creative Engineering, Kyonggi University, 94 San, Suwon-si 16227, Korea; modelan7@naver.com
* Correspondence: swchang@kyonggi.ac.kr; Tel.: +82-31-255-9739

**Abstract:** It is urgent to determine suitable municipal sludge treatment solutions to simultaneously minimize the environmental negative impacts and achieve sustainable energy benefits. In this study, different sludge pretreatment techniques were applied and investigated to enhance the sludge solubility and, subsequently, facilitate the anaerobic biodegradation performance of the mixed sludge under different sludge concentrations and pretreatment techniques. The sludge characteristics before and after pretreatment and batch experiments of anaerobic digestion of sludge samples under different conditions were analyzed and discussed. The results showed that the mechanical pretreatment method, alone and in combination with low-temperature heat treatment, significantly improved the sludge solubility, with the highest solubility at 39.23%. The maximum biomethane yield achieved was 0.43 $m^3$/kg after 10 d of anaerobic digestion of a 3% sludge sample subjected to mechanical and thermal pretreatment prior to anaerobic biodegradation. In comparison, it took more than 28 d to achieve the same biomethane production with the unpretreated sludge sample. Mechanical pretreatment and subsequent heat treatment showed a high ability to dissolve sludge and, subsequently, accelerate anaerobic digestion, thereby providing promising prospects for increasing the treatment capacity of existing and new sludge treatment plants.

**Keywords:** sewage sludge treatment; sludge solubilization; sludge pretreatment; anaerobic digestion; methane yield; mechanical pretreatment

## 1. Introduction

A large amount of biological sludge is generated daily from municipal wastewater treatment systems worldwide. In the Republic of Korea, 11,726 thousand t/d of sewage sludge was produced in 2019, an increase of 39.4% from 2009. As the percentage of the population with sewer system connection increases, sewage sludge production is forecasted to continue to increase [1]. Accordingly, plants that produce biogas from biomass waste in the Republic of Korea increased dramatically, from 90 plants (2016) to 98 plants (2017). Among them, plants that convert sewage sludge into biogas increased from 34 plants to 36 plants and are forecasted to continue to increase in the coming decades due to the environmental and economic benefits these systems provide [1].

According to the International Energy Agency, the global primary energy supply in 2016 was 13,761 million t (oil equivalent), with fossil fuel accounting for 81.1%. The Republic of Korea created a plan to produce new renewable energy using bioresources centering around advanced countries, such as European countries and Japan, and has begun increasing the amount of waste resources and biomass among new renewable energy sources used in Economic Cooperation and Development member countries. Consequently, it is expected that the energy content based on waste resources and biomass among

new renewable energy sources will continue to increase [2,3]. Sewage sludge containing biodegradable organic matter has received attention as a bioenergy source for sustainable development of the environment and society [4–7]. Reuse via bio gasification, composting, and solidification applies to sewage sludge, and the importance of the anaerobic digestion process, which can effectively enhance the value of resources, reduce pathogenic bacteria and malodor, and decrease sludge solid material, has been increasingly stressed [8,9].

Solubilization has been applied to induce the emission of organic matter contained in sewage sludge. Various methods, including thermal solubilization using heat exchangers, steam injection and autoclaving, chemical solubilization via chemicals, and mechanical solubilization via physical activity, have been reported [10–12]. The pretreatment method applied first to improve sludge digestion efficiency is thermal solubilization [13]. A previous study reported that thermal solubilization induces the emission of organic matter in a cell by decomposing the sludge gel structure, and the optimum temperature is 150–180 °C [14]. Among physical solubilization methods, ultrasonic wave pretreatment induces cavitation at a low frequency of 20–40 kHz for chemical reactions via the formation of OH, $HO_2$, and H. Application of a specific energy value of 1000–16,000 kJ/kg total solids (TS), according to the sludge TS concentration, is the most efficient method [15]. In addition to the abovementioned methods, liquid shear destroys the cell floc by mechanical force via high liquid flow from a high-pressure system. The breaker disc spray method (H.B. Choi) destroys the cell wall by spraying sludge liquid on a breaker disc at a speed of 30–100 m/s and undergoing rapid decompression after passing through a nozzle under the pressurization condition of 30–50 bar on a high-pressure pump. In addition, the ball-mill method solubilizes sludge by agitating a ball with a diameter of 0.25–0.35 mm at a speed of 6–10 m/s.

Examples of chemical solubilization are oxidation treatment and alkali treatment using KOH, NaOH, $Mg(OH)_2$, and $Ca(OH)_2$ [16]. The most widely used method for oxidation treatment is ozone treatment. However, a study reported that excessive use of ozone reduces the methane yield rate owing to the oxidation of soluble organic matter. A study on the optimum ozone input to improve sludge biodegradability was also conducted [17]. Alkali treatment is known to be the most effective chemical pretreatment method. A study reported that alkali treatment has an advantage because an increase reduces energy consumption in sludge temperature caused by the chemical reaction along with thermal solubilization. A study reported that using excessive alkali chemicals causes $Na^+$ and $K^+$ to accumulate, which acts as an inhibiting factor in the acid production and methane production reactions, thereby resulting in reduced methane production [18].

The abovementioned solubilization methods facilitate the hydrolysis of sludge via chemical and physical methods. Studies on pretreatment processes have been conducted to reduce sludge and improve biogas production, because the above methods are excellent at increasing dissolved organic matter, biodegradation, and biogas production. However, studies on mechanical solubilization are insufficient compared with studies on other solubilization methods, and there are still many aspects that need to be explored and elucidated, especially those on crushing solubilization techniques using shearing force, which can effectively solubilize sewage sludge by installing solubilization systems in the front of the anaerobic digestion process of new or existing sewage disposal plants; thus, studies on optimization are needed [15].

This study evaluated the applicability of a new method of crushing solubilization using shearing force by analyzing the shearing force in the physical crushing solubilization process, the changes in the physicochemical properties of the mixed sludge, and the biodegradability effect at the time of anaerobic digestion, which may depend on the solubilization duration of mixed sludge according to the TS concentration. This study also evaluated the combined pretreatment efficiency of thermal and physical solubilization by conducting thermal solubilization and combined pretreatment, which are known to effectively solubilize organic matter with extracellular high molecular substances and have many advantages [16] in terms of the initial equipment cost, chemical cost, and electric

power consumption. Specifically, high-temperature conditions in thermal solubilization require high energy consumption and stability. At a temperature of 170 °C or higher, insoluble substances are produced by the Maillard reaction and burnt sugar reaction, and toxic substances such as $NH_4^+$ are produced by chemical reaction, which reduce gas production when applying anaerobic digestion; thus, low-temperature thermal solubilization of less than 100 °C was applied [19–21].

In this study, we aimed to compare and evaluate the influence of sludge pretreatment techniques on the characteristics of sludge at different concentrations, the role of sludge pretreatment on the efficiency of anaerobic biodegradation processes, and the potential for biomethane production. This study will provide new directions for biological sludge treatment.

## 2. Materials and Methods

### 2.1. Substrate Preparation

The mixed sludge used in this study was a mixture of primary sludge and excess activated sludge, which was taken from S Wastewater Treatment Plant (S-WTP; 85,000 m$^3$/d), South Korea. After the sludge was collected, it was filtered through a 1 mm sieve to remove unwanted objects such as gravel and garbage. These two sludges were concentrated by evaporation to the same desired TS concentration, mixed together at a mass ratio of 1:1, stored at 4 °C to minimize variation in properties, and utilized throughout the experiments. The characteristics of the sludge mixture at different TS concentrations are presented in Table 1.

**Table 1.** Characteristic of mixed sludge at different total solids (TS) concentrations.

| Parameter | Unit | Mixed Sludge Class | | |
|---|---|---|---|---|
| | | Mixed Sludge 3% | Mixed Sludge 5% | Mixed Sludge 7% |
| TS | g/L | 28.3 | 59.6 | 81.1 |
| VS | g/L | 0.2 | 0.6 | 0.7 |
| TCOD$_{Cr}$ | g/L | 30.2 | 50.3 | 70.4 |
| SCOD$_{Cr}$ | g/L | 22.7 | 38.3 | 54.2 |
| TN | g/L | 2.5 | 4.3 | 6.5 |
| NH$_{4+}$ | g/L | 0.03 | 0.05 | 0.06 |
| pH | - | 7.32 | 7.28 | 7.34 |
| Alkalinity | mg/L as CaCO$_3$ | 300 | 450 | 600 |

### 2.2. Mechanical Solubilization and Operating Conditions

The mechanical sludge pretreatment (mechanical solubilization) applied in this study used hydro-mechanical shear forces to break and crush the sludge structures, i.e., to break down the microorganism cell walls and dissolve them into the soluble phase in the sludge liquid medium. The mechanical solubilization of mixed sludge was conducted by shearing force that occurred within a gap (0.2 mm) between the rotating rotor at the bottom of the crushing device and fixed stator using a high-shear mixer (model HM1HF, K&S Co., Ltd., Hwaseong, South Korea; Figure 1) to homogenize the sludge mixture. The mixed sludge dispersal process was conducted in batches, and each liter of mixed sludge was stirred and dispersed for 120 min at a rotational speed of 1000–4000 rpm.

### 2.3. Thermal Hydrolysis and Operating Conditions

After mechanical pretreatment (M), the sludge was further treated by thermal treatment at a low temperature (90 °C) to improve the hydrolysis capacity of the sludge, which was denoted as mechanical–thermal (MT) pretreatment. The thermal solubilization device used in this study had a total volume of 1.0 L (working volume of 0.7 L) and was designed as a compact and completely closed type to prevent errors in moisture content change due to the elevated temperature in the reactor. The reactor (pressured vessel) was made of stainless steel, and thus could operate stably under high temperatures and pressures

(Figure 2). This device was connected with a control panel, which could control parameters such as the stirring speed of the agitator, temperature, and pressure in the pressurized vessel. A circulating water pipeline system was designed to cool the pretreated sludge. More details on this device can be found in a previous publication by the authors [5].

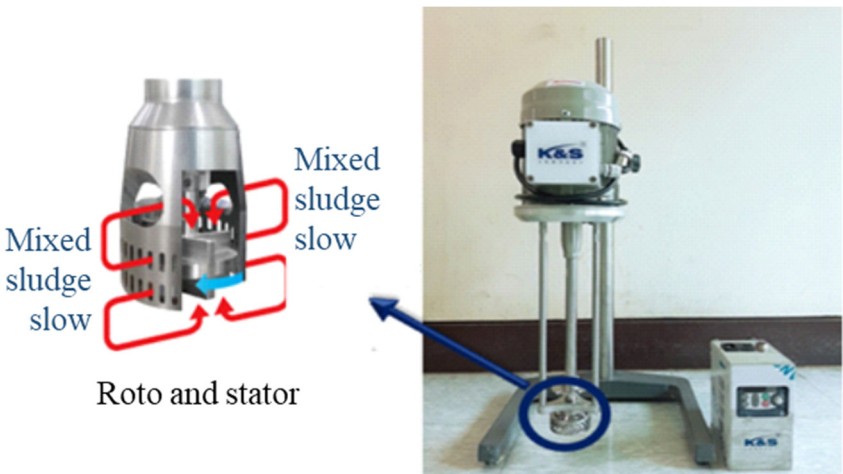

**Figure 1.** Mechanical pretreatment device.

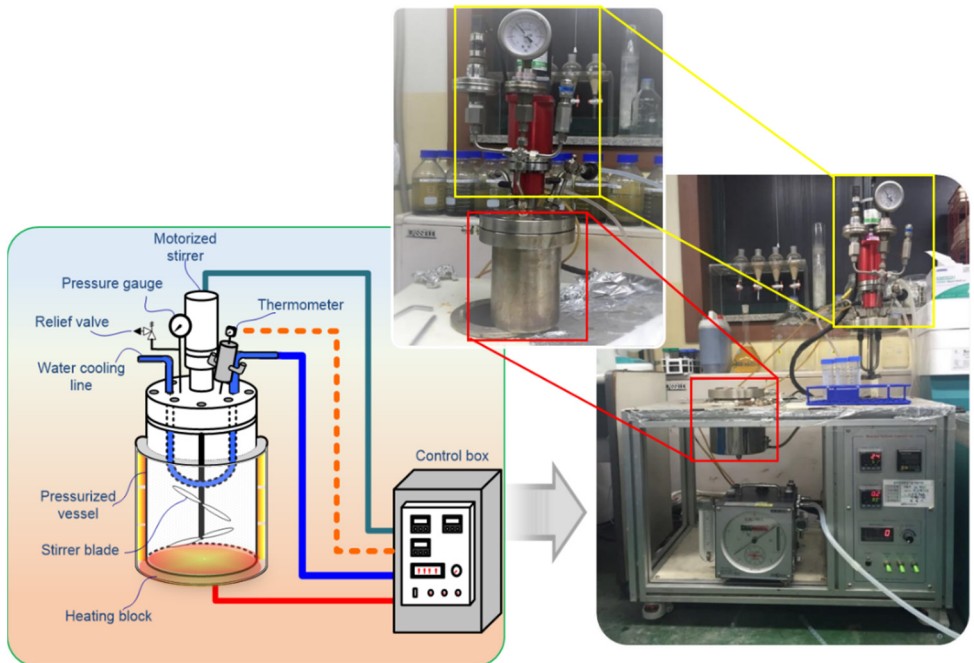

**Figure 2.** Thermo pretreatment device, schematic diagram, and photographs.

To evaluate the hydrolysis capacity of the sludge by pyrolysis at different TS concentrations, 0.7 L of the mixed sludge after physical crushing pretreatment was placed in the reaction vessel and operated at a temperature of 90 °C for 120 min.

### 2.4. Experimental Conditions and Analytical Methods

2.4.1. Characteristics of Solubilization Samples and Biogas Analysis Method

The symbols of the sludge samples and the corresponding pretreatment conditions are summarized in Table 2. The sample was mechanically pretreated, and the combination of mechanical and thermal conditions was denoted by MT. The characteristics of the sludge samples before and after pretreatment were analyzed and evaluated using the

parameters $TCOD_{Cr}$, $SCOD_{Cr}$, TS, volatile solids (VS), $NH_4^+$ and pH, according to the Standard Methods for the Examination of Water and Wastewater. The pH of the filtrate was measured using a pH meter (Hanna HI223, Smithfield, RI, USA). For dissolved substances of $SCOD_{Cr}$, $NH_4^+$ in the filtrate was analyzed and measured, which was obtained from sludge samples that were centrifuged at 6000 rpm and then filtered using a GF/C filter (Whatman, Maidstone, Kent, UK). The biochemical methane potential (BMP) was also examined, to compare and evaluate the highest biogas potential of pretreated substrates using different pretreatment techniques. The BMP test was performed by collecting a 500 μL sludge sample with a syringe (HA-81256, Hamilton Company, Reno, NV, USA) and measuring the sample using a thermal conductivity detector of a gas chromatograph (Agilent 7890A, Agilent Technologies, Inc., Santa Clara, CA, USA) equipped with a column (HP-PLOT/Q, Agilent Technologies, Inc., USA). The temperature conditions of the injector and detector were 230 and 250 °C, respectively. The oven temperature was set at 60 °C in the early stage and increased to 240 °C at a rate of 30 °C/min, and He gas was used as the carrier gas.

**Table 2.** Sample composition and pretreatment conditions employed in mechanical pretreatment and combined mechanical–thermal pretreatment.

| Samples | Mixed Sludge TS (%) | Pretreatment Methods and Conditions | | | |
| --- | --- | --- | --- | --- | --- |
| | | Speed (rpm) | Shear Rate (1/s) | Temperature (°C) | Time (min) |
| Mechanical pretreatment | | | | | |
| MS3% | 3 | - | - | - | - |
| 1S3% | 3 | 1000 | 17,166 | - | 120 |
| 2S3% | 3 | 2000 | 34,331 | - | 120 |
| 3S3% | 3 | 3000 | 51,497 | - | 120 |
| 4S3% | 3 | 4000 | 68,661 | - | 120 |
| MS5% | 5 | - | - | - | - |
| 1S5% | 5 | 1000 | 17,166 | - | 120 |
| 2S5% | 5 | 2000 | 34,331 | - | 120 |
| 3S5% | 5 | 3000 | 51,497 | - | 120 |
| 4S5% | 5 | 4000 | 68,661 | - | 120 |
| MS7% | 7 | - | - | - | - |
| 1S7% | 7 | 1000 | 17,166 | - | 120 |
| 2S7% | 7 | 2000 | 34,331 | - | 120 |
| 3S7% | 7 | 3000 | 51,497 | - | 120 |
| 4S7% | 7 | 4000 | 68,661 | - | 120 |
| Mechanical–thermal pretreatment | | | | | |
| MS3% | 3 | - | - | - | - |
| 1ST3% | 3 | 1000 | 17,166 | 90 | 120 |
| 2ST3% | 3 | 2000 | 34,331 | 90 | 120 |
| 3ST3% | 3 | 3000 | 51,497 | 90 | 120 |
| 4ST3% | 3 | 4000 | 68,661 | 90 | 120 |
| MS5% | 5 | - | - | - | - |
| 1ST5% | 5 | 1000 | 17,166 | 90 | 120 |
| 2ST5% | 5 | 2000 | 34,331 | 90 | 120 |
| 3ST5% | 5 | 3000 | 51,497 | 90 | 120 |
| 4ST5% | 5 | 4000 | 68,661 | 90 | 120 |
| MS7% | 7 | - | - | - | - |
| 1ST7% | 7 | 1000 | 17,166 | 90 | 120 |
| 2ST7% | 7 | 2000 | 34,331 | 90 | 120 |
| 3ST7% | 7 | 3000 | 51,497 | 90 | 120 |
| 4ST7% | 7 | 4000 | 68,661 | 90 | 120 |

### 2.4.2. Evaluation of Solubilization Efficiency

Sewage sludge solubilization efficiency refers to the efficiency of the conversion of particulate matter into dissolved substances after solubilization. Sewage sludge solubilization efficiency serves as an indicator for evaluating the efficiency of various solubilization methods and conditions. Sewage sludge solubilization efficiency is expressed as the percentage of particulate chemical oxygen demand ($PCOD_{Cr}$), which is the concentration of particulate organic matter before solubilization is applied, and $SCOD_{Cr}$ after solubilization was calculated according to Equation (1).

$$COD_{\text{Solubilization}}(\%) = \frac{COD_S - COD_{So}}{COD_{Po} - COD_{So}} \times 100 \tag{1}$$

where $COD_S$ is the $SCOD_{Cr}$ after solubilization (g/L), $COD_{So}$ is the $SCOD_{Cr}$ before solubilization (g/L), and $COD_{Po}$ is the $PCOD_{Cr}$ of the raw sludge before solubilization (g/L).

### 2.4.3. BMP Test Conditions

The BMP test was conducted to evaluate the anaerobic biodegradability of the mixed sludge and methane generation efficiency according to various conditions via technical solubilization. Digestion sludge from S-WTP was used as the seed sludge. The culture medium for supplying nutrients to anaerobic microorganisms in digestion sludge was modified according to the method presented by Shelton and Tiedje [2]. The components of the anaerobic culture medium are listed in Table 3.

**Table 3.** Anaerobic medium composition.

| Material | | Concentration |
| --- | --- | --- |
| Trace metals | $MnCl_2 \cdot 4H_2O$ | 0.50 mg/L |
| Trace metals | $H_3BO_3$ | 0.05 mg/L |
| Trace metals | $ZnCl_2$ | 0.05 mg/L |
| Trace metals | $CuCl_2$ | 0.03 mg/L |
| Trace metals | $NaMo_4 \cdot 2H_2O$ | 0.01 mg/L |
| Trace metals | $CoCl_2 \cdot 6H_2O$ | 0.50 mg/L |
| Trace metals | $NiCl_2 \cdot 6H_2O$ | 0.05 mg/L |
| Trace metals | $Na_2SeO_3$ | 0.05 mg/L |
| Mineral salts | $NH_4Cl$ | 0.530 g/L |
| Mineral salt | $CaCl_2 \cdot 2H_2O$ | 0.075 g/L |
| Mineral salt | $MgCl \cdot 6H_2O$ | 0.100 g/L |
| Mineral salt | $FeCl_2 \cdot 4H_2O$ | 0.020 g/L |
| Phosphate buffer | $KH_2PO_4$ | 0.270 g/L |
| Phosphate buffer | $K_2HPO_4$ | 0.350 g/L |

The culture medium produced was sterilized for approximately 15 min at 120 °C using an autoclave and then cooled at room temperature until a temperature of 35 °C was reached. Digestion sludge, which is equal to 1/10 of the total amount of culture medium, was injected to create seed sludge. Three hundred milliliters of seed sludge and solubilization sample was then injected into a 500 mL Duran bottle with a concentration of 2 g VS/L based on the seed sludge volume. Subsequently, 0.02 N HCl and 0.02 N NaOH were injected to adjust the pH of the inoculated mixed sludge and digestion sludge to neutral conditions. In addition, 1.2 g/L of $NaHCO_3$, which is an alkali substance, was injected to prevent pH reduction due to the acid production reaction in the early stage. Oxygen in the bottle was removed via $N_2$ gas purging at each stage of the BMP test seeding to create anaerobic conditions. After seeding was completed, the top of the reactor was sealed with a plastic cap made of rubber packing and a silicon cap. Finally, a test was conducted on the culture medium with a temperature of 35 °C. Biogas production was measured using 10 and 50 mL gas-tight syringes.

## 3. Results

### 3.1. Effects of Pretreatment Methods on Sludge Characteristics

Changes in the physicochemical properties of the sludge were evaluated by analyzing the sludge properties before and after mechanical pretreatment and MT pretreatment under different conditions. The changes in the characteristics of the solubilized sludge pretreated with different TS concentrations are shown in Figure 3. The experimental results showed that the TS, TCOD, and total N concentrations before and after pretreatment by mechanical and MT methods were not significantly changed, even when the shear rate increased from 17.2 to 68.7 1/s of mechanical pretreatment. Meanwhile, the pH of the soluble substrate tended to decrease slightly from $7.33 \pm 0.04$ to $6.37 \pm 0.04$, corresponding to the increase in the rotation speed of the pretreatment device from 1000 to 4000 rpm, which could be explained by the solubilization of particulate organic matter in EPS [22]. In contrast, $SCOD_{Cr}$ and $NH_4^+$ significantly increased after pretreatment and increased proportionally to the rotation speed of the mechanical pretreatment device; specifically, the $SCOD_{Cr}$ and $NH_4^+$ concentrations after mechanical pretreatment increased by $21.32 \pm 2.61$ and $10.11 \pm 0.67\%$, respectively. When the sludge sample was subjected to further thermal pretreatment at 90 °C for 2 h, the $SCOD_{Cr}$ and $NH_4^+$ concentrations increased to $35.42 \pm 4.01$ and $14.87 \pm 1.21\%$, respectively.

The results also showed that, under the same experimental conditions, the fraction of COD solubility was higher at lower sludge concentrations; for example, under a mechanical pretreatment condition of 4000 rpm with a sludge concentration of 3% (TS3%), the COD dissolution rate was 24.50%, whereas under the same experimental conditions with a sludge concentration of 7% (TS7%), the COD dissolution rate was only 18.11%. The $NH_4^+$ concentration after dissolution did not show a significant difference between the different sludge concentrations applied and fluctuated in the range of 9.46–11.03% under a mechanical pretreatment condition of 4000 rpm. After these sludge samples were disintegrated by the mechanical method and further thermally pretreated at a low temperature of 90 °C for 2 h to continue the hydrolysis of these sludge samples, the disintegration efficiency of the sludge continued to increase by 1.60–1.72 times for SCOD and 1.34–1.64 times for $NH_4^+$. Thus, it can be concluded that the solubility of the sludge depends on the sludge concentration and shear stress of the mechanical pretreatment device. The mechanical sludge dissolution method applied in this study achieved the highest efficiency when operating at a rotational speed of 4000 rpm and a sludge concentration of 3–5%. To achieve a higher sludge disintegration efficiency, it is necessary to continue thermal treatment at a low temperature of 90 °C.

In addition, Wilson and Novak [23] reported that the $NH_4^+$ concentration in the anaerobic digester of sewage sludge was approximately 0.8 g/L and did not clearly inhibit the anaerobic degradation processes. Meanwhile, Sung and Liu (2003) reported that when the $NH_4^+$ concentration of the digestion sludge increased to 5.77 g/L in the anaerobic digester, anaerobic microbial activity was inhibited, and methane production decreased by up to 64%. Thus, the $NH_4^+$ concentration achieved in all the disintegrated sludge samples was within the allowable limits for stable anaerobic biodegradation processes.

### 3.2. Effect of Pretreatment on Mixed Sludge Solubilization Efficiency

The studied sludge mixture mainly consisted of organic matter, had low solubility, and had low and stable VS/TS and SCOD/TCOD ratios of 0.752–0.768 and 0.707–0.770, respectively. When mechanical pretreatment of sludge was applied, the solubilization efficiency of all the samples pretreated under the same sludge concentration conditions increased proportionally to the shear stress and increased significantly when the shear stress was increased from 17,166 to 68,661 1/s (Figure 4). For mixed sludge samples with TS concentrations of 3%, 5% and 7%, which were mechanically pretreated under a shear stress condition of 6866 1/s, the sludge solubilization efficiency increased significantly by 22.4%, 20.8% and 18.3%, respectively; however, there was no significant difference in the solubilization efficiency between these samples. Meanwhile, the sludge solubilization

efficiency of sludge samples that were mechanically pretreated and subsequently thermally treated at a low temperature of 90 °C was significantly improved and was 1.68 ± 0.04 times higher than that of mechanical pretreatment alone, with increases of 39.23% for TS3%, 33.60% for TS5% and 31.0% for TS7% (Figure 4). Thus, the results indicated that the higher the sludge concentration, the lower the sludge solubility of the two applied pretreatment methods. It is assumed that a higher sludge concentration will increase the viscosity of the sludge, thereby limiting the heat transfer and mass transfer in the sludge medium. However, thermal treatment is very important in improving the sludge dissolution efficiency, and it seemed that thermal treatment increased the contents of soluble proteins and intercellular organics and decreased the viscosity of the mixed sludge.

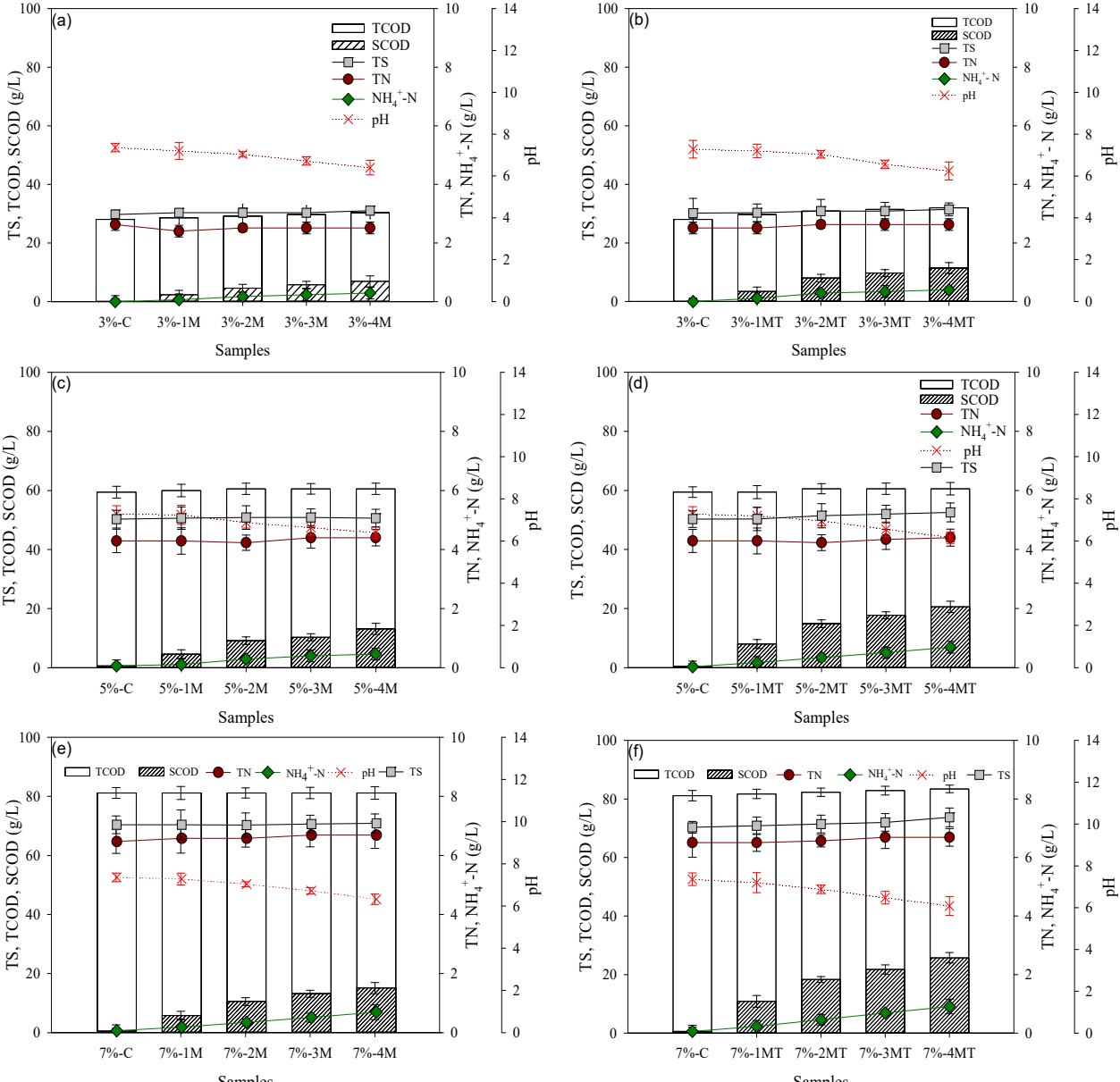

**Figure 3.** Influence of the sludge concentration (TS3, TS5, and TS7%) and pretreatment techniques (mechanical pretreatment (**a**,**c**,**e**) and mechanical–thermal pretreatment (**b**,**d**,**f**)) on sludge characteristics; error bars are the standard error of the data.

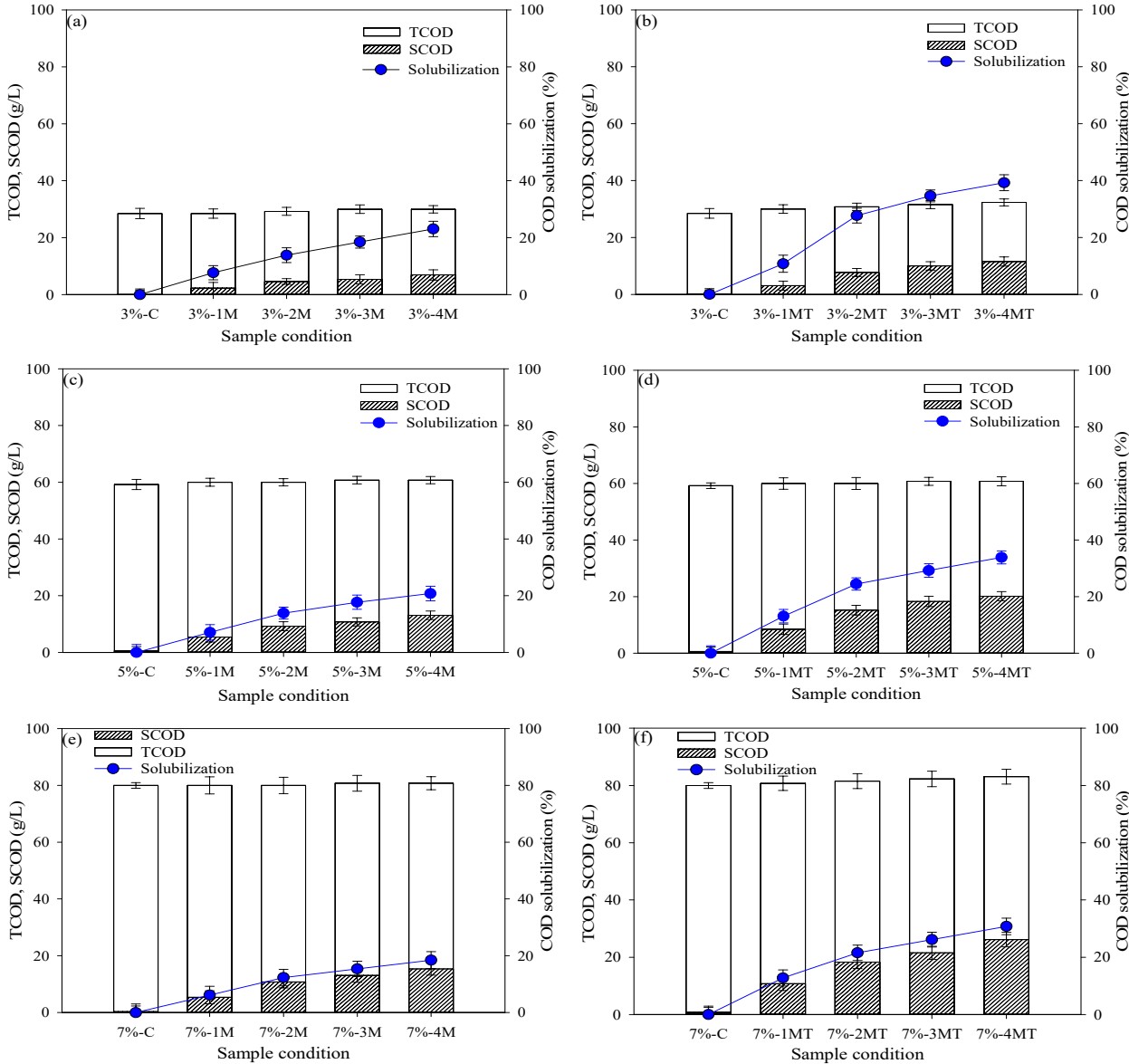

**Figure 4.** Influence of mechanical pretreatment (**a**,**c**,**e**) and mechanical–thermal pretreatment (**b**,**d**,**f**) on solubilization properties of sludge with different concentrations (TS3, TS5, and TS7%); error bars are the standard error of the data.

Thus, the research results showed that the combination of mechanical pretreatment and subsequent thermal treatment will improve the solubilization efficiency of organic matter in the sludge more than mechanical pretreatment alone. Previous studies have also shown the important role of thermal pretreatment in improving the sludge disintegration efficiency; however, these studies were conducted at high temperatures, thereby leading to high energy consumption. Under optimal conditions, remarkable activated sludge solubilization efficiencies have been achieved. For example, a solubilization efficiency of 25.3% was achieved at 130 °C by Valo et al. [24], 18.0% was achieved at 121 °C by Kim, Park, Kim, Lee, Kim, Kim and Lee [16], and 41.0% was obtained with a mixture of activated sludge (20.0%) and wine effluent (80.0%) at 170 °C by Bougrier et al. [25] (Table 4).

**Table 4.** Solubilization of sludge under different pretreatment techniques.

| Substrate | Pretreatment | TS (%) | Conditions | Solubilization (%) | Reference |
|---|---|---|---|---|---|
| Mixed sludge | Mechanical | 3.0 | 4000 rpm, 2 h | 24.50% | This study |
| Mixed sludge | Mechanical–thermal | 3.0 | 4000 rpm, 2 h 90 °C, 2 h | 39.23% | This study |
| Activated sludge | Thermal | 1.7 | 130 °C | 25.00% | [24] |
| Activated sludge | Thermal | 3.8 | 121 °C, 30 min | 18.00% | [16] |
| Wine sludge (80%) Activated sludge (20%) | Thermal | 4.0 | 170 °C, 30 min | 41.00% | [25] |
| Activated sludge | Chemical | 1.3 | 1.5 g NaOH/L (pH 12, 30 min) | 18.00% | [26] |
| Activated sludge | Ultrasonic | 1.9 | 10,000 kJ/kg | 32.00% | [27] |

The evaluation of sludge solubilization with different TS concentrations revealed that, as the TS concentration increased, the maximum solubilization efficiency decreased. This might have occurred because highly concentrated mixed sludge is more vulnerable to crushing than less concentrated mixed sludge, owing to its higher viscosity and lower heat transfer coefficient at the time of thermal pretreatment. Accordingly, highly concentrated mixed sludge requires more energy consumption and a longer application time when mechanical and thermal pretreatments are applied [5].

### 3.3. Effects of Pretreatment Techniques on Anaerobic Biodegradability and Biomethane Production

According to the experimental results of the sludge pretreatment methods at different TS concentrations, nine sludge samples were selected to further evaluate the anaerobic biodegradability and biomethane generation via BMP tests, namely three raw sludge (untreated sludge) samples with TS concentrations of 3, 5 and 7% (named RS3, RS5 and RS7%, respectively), three sludge samples (TS3, TS5 and TS7%) mechanically treated with shear stress of 68,661 1/s (4S3, 4S5 and 4S7%, respectively), and three sludge samples (TS3, TS5 and TS7%) simultaneously mechanically treated and heat treated (4ST3, 4ST5 and 4ST7%, respectively). The results of the anaerobic digestion test presented in this section can lead to several important conclusions, such as which pretreatment method and sludge concentration are optimal for the greatest digester efficiency and highest biomethane yield. The results can also predict the time required for the biodegradation phases in the reactors (lag phase, logarithmic phase, and stationary and death phase) corresponding to the substrates [28].

The BMP test results for the substrates are shown in Figure 5. The results showed that all the pretreated samples had a faster biodegradation rate, and the ultimate methane yield was higher than that of the untreated sludge. The time to achieve final (maximum) methane production for sludge samples with pretreatment intervention was much shorter, i.e., approximately 13 d for mechanically and thermally pretreated sludge samples and 15 d for the sludge sample with mechanical pretreatment only, whereas the non-pretreated sludge sample took more than 27 d to achieve a maximum biogas yield of $0.425 \pm 0.018$ m$^3$/kg VS.

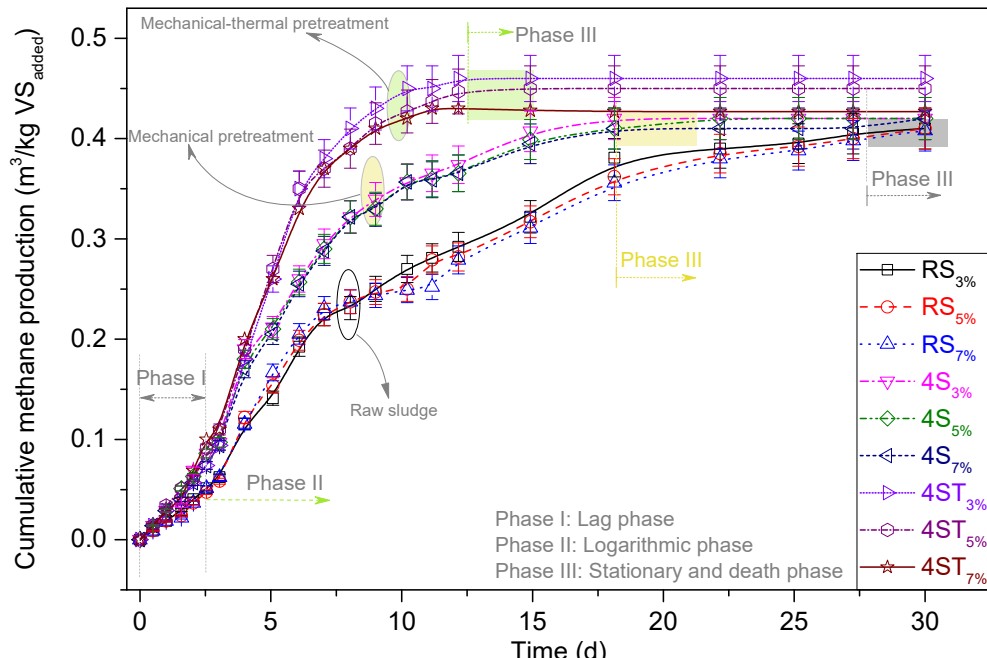

**Figure 5.** Cumulative biogas yield from different substrates tested (error bar are the standard error of the data).

The results shown in Figure 5 also indicated that, during anaerobic digestion, the lag time (phase I) for methane production of all sludge samples was not significantly different. However, when entering the growth stage (phase II), there was a very clear difference between samples. For untreated sludge samples, this period took more than 25 d to reach equilibrium, whereas for sludge samples that were pretreated by MT and mechanical treatment alone, this period took 11–12 and 17–18 d, respectively. Furthermore, for the range of sludge concentrations studied (TS 3–7%), anaerobic digestion of sludge samples with lower TS concentrations resulted in higher biomethane yields and a shorter time for anaerobic biodegradation to reach steady state.

Therefore, for sludge samples with pretreatment intervention, the biodegradation time was significantly reduced, but the maximum methane production was still achieved. Sludge samples with pretreatment intervention created a favorable environment for anaerobic microorganisms to adapt and grow faster. Moreover, the treatment of sludge samples with TS concentrations of 3–5% was more effective.

The combined MT pretreatment method of sludge yielded remarkable results, such as a high sludge solubilization efficiency leading to favorable conditions for faster biodegradation processes and higher methane production due to the decomposition of more compounds in the sludge. However, a study on the energy requirements and the relationship between sludge characteristics (carbohydrates and proteins), biodegradation time, and methane yield must be conducted to make predictions and design parameters for a complete sludge treatment system.

## 4. Conclusions

In this study, different sludge pretreatment techniques were investigated and evaluated. The combination of mechanical sludge pretreatment and subsequent low heat treatment showed better sludge dissolution efficiency; specifically, the dissolved COD content was 39.2% greater than that of the untreated sludge sample. The BMP test results showed that the pretreated sludge samples achieved a higher final biomethane yield in a shorter decomposition time, with a maximum methane yield of $0.43 \pm 0.016$ m$^3$/kg vs. over a 10.2 d period for the MT pretreated sludge sample. Meanwhile, the non-pretreated sludge sample required >29 d to achieve the same yield. The optimal sludge concentration

for maximum efficiency is suggested to be 3–5% TS. Although the combined pretreatment techniques showed great advantages in improving the efficiency of anaerobic digestion and methane production, to achieve widespread application, it is necessary to further study the relationship between parameters such as pretreatment time, sludge characteristics, gas yield and cost.

**Author Contributions:** Conceptualization, J.-Y.A.; methodology, J.-Y.A.; software, J.-Y.A.; validation, J.-Y.A.; formal analysis, J.-Y.A.; investigation, S.-W.C.; resources, J.-Y.A.; data curation, J.-Y.A.; writing—original draft preparation, J.-Y.A.; writing—review and editing, S.-W.C., J.-Y.A.; supervision, S.-W.C.; project administration, S.-W.C.; funding acquisition, S.-W.C. All authors have read and agreed to the published version of the manuscript.

**Funding:** This research received no external funding.

**Data Availability Statement:** The data used to support the findings of this study are available from the first author upon request.

**Acknowledgments:** This work was supported by Kyonggi University's Graduate Research Assistantship 2021.

**Conflicts of Interest:** The authors declare no conflict of interest.

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
