# Peer review of "Effects of Sludge Concentration and Disintegration/Solubilization Pretreatment Methods on Increasing Anaerobic Biodegradation Efficiency and Biogas Production"

_sustainability, doi:10.3390/su132212887_

Round 1
Reviewer 1 Report
The main aim of this study is to evaluate the influence of different sludge pretreatment techniques on the characteristics of sludge, the role of sludge pretreatment on the efficiency of anaerobic biodegradation processes, and the potential for biomethane production.
- The novelty of this study must more be highlighted in the introduction section.
- English used in this paper must be carefully polished.
- Error bars or standard deviation must be added to all Figures and Tables.
- The following important papers should be cited in this article: 10.15017/1906408; 10.15017/1960668; 10.1016/j.chemosphere.2021.131990.
Reviewer 2 Report
Dear Sir,
Greetings!
The submitted manuscript "Effects of sludge concentration and sludge disintegration and solubilization pretreatment methods on increasing anaerobic biodegradation efficiency and biogas yield". The paper is too general, not suitable as per my opinion. Please highlight the novelty of work in right manner. Clearly discuss outcomes and what research gaps it covers. Please show how this paper has a strong correlation with sustainability concerns. 
Grammar and syntax must be improved. Revision with the help of a native English speaker is highly recommended.
Thanks
Round 2
Reviewer 1 Report
- The following important papers should be cited in this article: 10.15017/1906408; 10.15017/1960668; 10.1016/j.chemosphere.2021.131990.
Reviewer 2 Report
Dear sir,
Greetings of the Day.
The author has finally corrected the manuscript as per correction needed. I also recommended this manuscript for the final publication.
Thanks for your invitation to review the manuscript
With Regards
Hanuman Singh Jatav